# Shear Performance Study of Sleeved Stud Connectors in Continuous Composite Girder

**DOI:** 10.3390/ma17133326

**Published:** 2024-07-05

**Authors:** Fei Wu, Hang Su, Qingtian Su, Bo Yuan

**Affiliations:** 1Department of Bridge Engineering, Tongji University, 1239 Siping Road, Shanghai 200092, China; 1910392@tongji.edu.cn; 2Shanghai Municipal Safety and Quality Supervision Administration for Construction Engineering, Shanghai 200032, China; suhangajzj@163.com; 3Shanghai Engineering Research Center of High Performance Composite Bridge, Shanghai 200092, China; 4Henan Provincial Communications Planning & Design Institute Co., Ltd., Zhengzhou 451450, China; yuanbo_xdp@sina.com

**Keywords:** connectors, composite beams, sleeve, welded studs, experimental study, finite element analysis

## Abstract

In order to reveal the mechanism of sleeved stud connectors, 15 push-out specimens were designed, and static loading tests were conducted to evaluate the mechanical performance. The shear performance differences between the novel sleeved studs and conventional welded studs were compared. Referring to the experimental results, an Abaqus nonlinear finite element model was established to study the shear mechanism of sleeved stud connectors. Parametric analysis was conducted to investigate the effects of stud height, sleeve filling material, and sleeve diameter on the mechanical performance of the connectors. The experimental and finite element analysis results indicated that the ultimate shear bearing capacity and shear stiffness of the sleeved stud connectors were higher than those of ordinary welded studs, and the maximum slip was relatively small. Compared to conventional welded studs, the ultimate bearing capacity of sleeved studs increased by 4% to 8%, and the shear stiffness increased by 25% to 35%. Since the shear behavior of sleeved studs mainly occurred at the base of the studs, the influence of stud height on shear performance was relatively small. However, sleeve and stud diameter have a great influence on bearing capacity and stiffness. As the Ultra-High Performance Concrete (UHPC) near the base of the stud effectively enhanced the shear carrying capacity of the sleeved stud connectors, the shear carrying capacity and shear stiffness increased with the increase in the sleeve diameter.

## 1. Introduction

Composite beam bridges are widely adopted as they combine the benefits of steel and concrete to create efficient bridge structures [1,2]. However, in continuous composite beam bridges, the cracking of the concrete deck in negative bending moment regions near supports is an issue [3,4]. Scholars have proposed several methods to prevent concrete cracking in the negative bending moment regions of composite bridges. For instance, increasing the reinforcement ratio [5,6] and adopting fiber-reinforced plastic [7,8] to the concrete slab can prevent concrete cracking. Adopting double-composite action can increase the cracking moment [9]. In addition, partial shear connections can also decrease concrete stress in the hogging moment region [10].

The post-tensioned composite beam bridge is also a common solution for cracking in negative bending moment regions of composite beam bridges [11], which involves prestressing the concrete bridge deck in the negative bending moment zone by post-tensioning. The construction of post-tensioned composite beam bridges must satisfy the fundamental principles of the post-tensioning method. That is, the concrete slab remains separated from the steel girder before applying prestress. After tensioning the prestressing tendons, the concrete slab is then integrated with the steel girder. Therefore, the connectors between the steel girder and concrete deck are a key structural detail.

Whether using cast-in-place bridge decks or prefabricated bridge decks, stud connectors are arranged in clusters, and corresponding stud holes should be reserved in the bridge deck. The cluster stud connector was first introduced in Switzerland [12], and numerous scholars have studied the mechanical performance of cluster stud connectors through experiments and finite element methods.

Shim et al. [13,14] conducted static and fatigue tests on push-out specimens to study the ultimate bearing capacity and fatigue performance of the cluster stud connection between prefabricated concrete bridge decks and steel beams. Ryu et al. [15,16] investigated the crack resistance of prefabricated concrete slabs through full-scale model tests and compared the cracked composite beam section bending stiffness with the formula recommended in Eurocode 4-2. Their study indicated that the crack width at the interface of prefabricated concrete slabs was greater than the specified formula value, and the spacing of transverse steel bars in the concrete slab significantly affected the crack spacing.

Okubo [17], Hou Wenqi [18], Liu Yongjian [19], and Xiang Yiqiang [20] conducted push-out tests to investigate the shear stiffness and shear bearing capacity of cluster stud connectors. The studies found that the cluster effect increased the non-linear stress of the cluster stud, and the average ultimate bearing capacity of the cluster stud was significantly reduced compared to the ultimate bearing capacity of a single stud. Wang et al. [21] conducted push-out tests and compared the shear-slip curves and failure modes of cluster studs with different stud hole shapes (circular/square), load paths (monotonic loading/cyclic loading), and casting methods (prefabrication/cast-in-place). Xu et al. [22,23,24,25] studied the mechanical performance of cluster stud connectors under the coupling effect of bending and shear through static push-out tests and finite element parameter analysis. They analyzed the reduction in shear stiffness and shear bearing capacity of the cluster stud caused by transverse bending moments leading to concrete cracking. In addition, fatigue performance under low-cycle loading was studied through cyclic loading tests, revealing fatigue damage near the root of the studs.

In the case of post-tensioned composite beams with cast-in situ bridge decks, a common practice on construction sites is to use single-use wooden formwork as the inner mold for cluster stud holes because of its construction economy. Such formwork is placed within the holes before tensioning the prestressing tendons, and it is later removed after the tendons have been tensioned. However, practical experiences have shown that removing the formwork from cluster stud holes on-site consumes a significant amount of time and labor. Moreover, due to the presence of longitudinal and transverse steel bars within the cluster stud holes, it is challenging to completely clean the wooden formwork, affecting the construction quality of the cluster stud holes. To address this issue, researchers have proposed two types of inner molds for cluster stud holes that can be directly embedded in the cast-in-place bridge deck: the encased cluster stud connector and the sleeved cluster stud connector. Both of these special cluster stud connectors adhere to the fundamental principles of the post-tensioning method, eliminating the need to extract inner molds and providing the convenience of standardized prefabrication in a factory setting.

The construction method for the encased cluster stud connector involves enclosing a thin steel plate to form a steel casing around the perimeter of the cluster studs before pouring the concrete bridge deck. The reinforcements pass through reserved holes in the steel casing, as illustrated in Figure 1a. The steel casing, serving as the inner mold of the bridge deck for the cluster stud holes, not only prevents the bonding of the steel beam with the concrete during the first casting process, but also prevents collisions between the studs and transverse steel bars. Embedded as a precast component in the concrete bridge deck, the steel casing has minimal impact on cluster studs. However, during the tensioning of prestressing tendons, the shape and spacing of the steel casing significantly influence the bridge deck. The weakening effect of the steel casing on the bridge deck results in an uneven distribution of prestressing stress, preventing uniform transfer of prestressing stress to the concrete between two longitudinal cluster stud holes.

To mitigate the weakening effect on the bridge deck, researchers have further proposed to use steel sleeves, in replacement of steel casings which have sharp rectangular shapes and a substantial cavity. In this construction approach, a corrugated steel sleeve is fitted over the welded stud as the inner mold for bridge deck concrete casting, as shown in Figure 1b. This design prevents the concrete from bonding with the studs during the casting process and maintains separation from the studs. Leakage prevention measures are required during the installation of the steel sleeves. The steel sleeve is filled with concrete after prestressing to establish a connection between the concrete slab and the welded studs, thus the steel main girder.

This study entails experiments and numerical investigations on the sleeved stud connectors, exploring their mechanical performance and failure mechanisms. Through parametric analysis, the study reveals the influence of stud dimension and filling material on the shear performance of the sleeved stud connectors.

## 2. Shear Test of Sleeved Stud Connectors

### 2.1. Specimen Design

The experiment designed 15 push-out specimens, and the basic characteristics of the specimens are shown in Table 1, with the configuration depicted in Figure 2 and Figure 3. In the specimen label, the letters “SS” represent the proposed sleeved stud, the letters “SN” represent the conventional welded stud, and the numerical ending represents the arrangement of studs on each side of the specimen. Specimens SN12, SS12, SN22, and SS22 are small push-out specimens with identical external dimensions and reinforcement layouts, differing only in the type and arrangement of connectors. SN22 and SS22 are cluster stud specimens with two rows and two columns of studs arranged on one side, with both longitudinal and transverse spacings between the studs set at 120 mm. Testing of these small push-out specimens provides insights into the shear performance differences between the proposed and traditional connectors. Specimens SSL12 to SSL52 are large push-out specimens, with identical external dimensions and reinforcement layouts, differing only in the arrangement of connectors. These specimens have a symmetrical arrangement of two columns of studs, increasing vertically from one row to five rows, with longitudinal and transverse spacings of the studs set at 120 mm. Testing of these large push-out specimens investigates the impact of the arrangement of the sleeved stud connectors on shear performance, allowing for an understanding of the cluster stud reduction effect of the connectors.

All specimens used C50 concrete, and the reinforcement employed was HRB400 with a diameter of 20 mm. The sleeves utilized metal corrugated sleeves with a diameter of 6 cm, a thickness of 0.35 mm, and a height of 320 mm. The studs uniformly used ML15 cylindrical head studs with a diameter of 22 mm and a height of 200 mm. The grouting material inside the sleeves was UHPC.

### 2.2. Loading Scheme

The loading was carried out using a shear testing machine, with a load applied at the steel structure. A rubber pad between the machine and the reaction frame ensures the evenly distributed load. Prior to the formal loading, the specimens underwent three preloads at 20% of the estimated load. During the formal loading, 60% of the estimated load was set as the demarcation point. In the first phase, the loading rate is 5 kN/s, and in the second phase, the loading rate is 0.5 mm/min. The frequency of the measurement system was set at 2 Hz.

Four displacement sensors were applied to specimens with a single row of studs. For specimens with multiple rows of studs, eight displacement sensors were placed around the steel-concrete interface. Sensors numbered 1 to 4 were positioned at the first row of studs, and sensors numbered 5 to 8 were placed at the last row of studs. The numbering sequence followed the same order as in specimens with a single row of studs. The layout of measurement points and the loading device are illustrated in Figure 4.

### 2.3. Material Properties

This experiment conducted tests on the compressive strength, axial compressive strength and elastic modulus of cubic specimens made of C50 ordinary concrete and Ultra-High Performance Concrete (UHPC) according to GB/T 50081-2019 [26]. Additionally, the yield strength, ultimate tensile strength, and elastic modulus of the studs were also tested according to GB/T 228.1 [27]. The test results are presented in Table 2.

## 3. Experimental Results and Analysis

### 3.1. Failure Modes

The typical failure modes of both conventional welded stud push-out specimens and sleeved stud push-out specimens are illustrated in Figure 5. In all specimens, the studs experienced shear failure at their bases, with localized concrete compression damage occurring beneath the base of the studs. It can be found in Figure 5 that the steel sleeve beneath the studs was also crushed. Due to the higher strength of the UHPC at the base of the studs, the concrete failure zone of the sleeved studs was smaller than conventional welded studs. This suggests that, during the testing process, the sleeved studs primarily underwent direct shear at the base, while the conventional welded studs exhibited more pronounced bending effects. As the major loading position of the sleeved stud is at the root, the influence of stud height on shear performance was minimal.

### 3.2. Load-Displacement Curve

The load-slip curves for the push-out tests are presented in Figure 6, where the horizontal and vertical axes, respectively, represent the average slip and the average shear load per stud. The load-slip curves for the sleeved studs exhibit slightly higher variability compared to conventional welded studs. This was primarily attributed to the impact on the metal corrugated sleeves during concrete casting, causing difficulty in keeping the studs centered within the sleeves.

Figure 6c distinctly illustrates the shear performance differences. The sleeved studs exhibited higher shear bearing capacity and stiffness than the conventional welded studs, with a relatively smaller slip at the onset of ultimate bearing capacity.

The impact of the arrangement differences of the sleeved stud connectors on shear performance is demonstrated in Figure 6d. As the number of rows of sleeved studs increased, there was a general trend of decreasing ultimate bearing capacity and shear stiffness of the studs. Notably, specimen SSL32, with sleeved studs, experienced grout leakage, leading to a lower ultimate bearing capacity. This highlighted the significance of construction quality for the shear performance of sleeved studs.

### 3.3. Analysis of Test Results

The summarized test results for each specimen are presented in Table 3, where $V_{u}$ represents the ultimate shear bearing capacity of the studs, $S_{p}$ represents the slip corresponding to the ultimate shear bearing capacity, and $K_{0.2\;mm}$, $K_{1/3}$, and $K_{1/2}$ represent the shear stiffness of the studs, determined at slip values corresponding to 0.2 mm, 1/3$V_{u}$, and 1/2$V_{u}$, respectively. Table 3 outlines the variations in shear performance for each specimen group, with the results for small push-out specimens normalized to the SN12 specimen and for large push-out specimens normalized to the SSL12 specimen.

The ultimate shear bearing capacity of the sleeved studs increased by 4.5% compared to conventional welded studs, with a reduction of 27.7% in ultimate slip. This indicated that the sleeved studs showed an insignificant advantage in improving shear strength, but they exhibited enhanced resistance to slip deformation. The shear stiffness $K_{0.2\;mm}$ of the sleeved studs increased by 31.9% to 35.6% compared to conventional welded studs. Additionally, by comparing specimens SS22 and SN22, it was evident that the sleeved studs were less influenced by the cluster stud effect than conventional welded studs, indicating that the shear performance of the steel sleeve cluster stud connector was superior to that of the conventional cluster stud connector.

To investigate the cluster stud effect of the sleeved studs, specimens SSL12 to SSL52 were compared. Overall, the shear strength decreased with an increase in the number of rows of studs, similar to the results for small push-out specimens. The difference between SSL12 and SSL22 was not substantial, but the shear strength and stiffness of the studs significantly decreased after adding the third row of studs. Due to the uncertainty in the position of the sleeves relative to the studs, there is considerable variability in the plastic phase, and the ultimate shear bearing capacity exhibited no clear regularity. From specimens SSL22 to SSL52, the shear stiffness $K_{0.2\;mm}$ decreased by 5.5%, 11.1%, 20.9%, and 25.0%, indicating that the average shear stiffness of multiple rows of studs was lower than that of a single row of studs.

## 4. Numerical Simulation

### 4.1. Modeling Methods

In order to further analyze the shear performance of the sleeved studs, Abaqus (Version 6.3) finite element (FE) models were established based on the test specimens. Considering the symmetry of the specimens, only a quarter of the FE model was built, as shown in Figure 7. The model included steel sleeves, concrete blocks, steel structures, reinforcements, and welded studs. The concrete block was composed of UHPC and ordinary concrete. Simultaneously, reference points were set at the center of a rigid base to obtain the reaction forces (a 10 mm loading displacement).

The concrete blocks, welded studs, and steel structures of the specimen were modeled using C3D8R solid elements, while the reinforcement used T3D2 truss elements. The overall element size of the specimen was 15 mm, while the concrete elements around the welded studs and steel structures were locally refined to 3 mm.

The model employed surface-to-surface contact simulation for the interfaces of steel-concrete bonding surfaces, C50-UHPC, and UHPC-welded studs. Tangential contact was simulated using the penalty function friction formula, and the friction coefficient was set as 0.1 [28]. Normal contact was simulated using hard contact. Constraints between steel reinforcement and concrete were implemented using the embedded command, and tie constraints were applied between the rigid bases and the contacting surfaces. Since the model represented only a quarter of the actual test specimen, appropriate constraints were applied to the symmetric planes. Additionally, the model did not consider the frictional effect between the bottom surface of the concrete block and the loading device; it assumed complete fixation of the bottom surface of the concrete block to the loading device.

The simulation employed Dynamic/Explicit method for analysis. The target time increment of the mass scaling was set at 0.0001 to improve the calculation efficiency.

### 4.2. Material Constitutive Models

(1)C50 Concrete Constitutive Model

The stress–strain relationship of concrete is simulated using the concrete damage plastic model. This model employs the plastic damage factor (d) to quantify the stiffness degradation. The damage factor in the CDP model is based on the energy equivalence principle proposed by Sidoroff [29], and its expression is given in Equation (1).
(1)$$d=1-\sqrt{\frac{\sigma }{E_{s}\varepsilon }}$$

Constitutive model for C50 concrete was calculated through the following steps. Firstly, based on material tests, the initial elastic modulus $E_{s}=$ 35.77 GPa and the axial compressive strength $f_{c,r}=$ 39.9 MPa were determined. Following the methodology specified in GB 50010 [30], the limit compressive strain $\varepsilon_{c,r}=$ 0.0017 was calculated, and the damage factors ($d_{c}$) for different stages were obtained, resulting in the stress–strain (σ−ε) curve, as shown in Figure 8. Subsequently, the data were input in two segments. First segment required only the elastic modulus and compressive strength, while the nonlinear segment necessitated the computation of non-elastic strains ${\tilde{\varepsilon }}_{c}^{in}$ for each stage (or crack opening strain ${\tilde{\varepsilon }}_{t}^{ck}$ for tensile), and the corresponding damage factor was calculated using the Sidoroff energy equivalence principle.

(2)UHPC Constitutive Model

The constitutive model of UHPC was similar to C50 (as shown in Figure 9). For the compressive constitutive model, the axial compressive strength ($f_{c}$) was experimentally determined as 129.4 MPa. The limit compressive strain ($\varepsilon_{c.r}$) was set to 0.0035, and the initial elastic modulus ($E_{c}$) was determined based on material test results with a value of 44.0 GPa, while $E_{s}$ represented the secant modulus at the stress peak point. For the tensile constitutive model, $f_{ct}$ was set to 8.0 MPa, and the peak strain ($\varepsilon_{ca}$) and ultimate strain ($\varepsilon_{pc}$) were assigned values of 0.0002 and 0.002, respectively. The parameter α primarily influences the descending segment and was determined based on the value of the steel fiber parameter, set to 1.106.

(3)Steel Constitutive Model

The stress–strain curve for the material of the welded studs followed a three-segment model, as depicted in Figure 10a. Yield and ultimate stress were initially determined by material testing, which were respectively 368 MPa and 498 MPa. The ultimate strain was set at 2.5%, and the elastic modulus was assigned a value of 207 GPa. The stress–strain curves for both steel plates and reinforcements were modeled using a two-segment model, as illustrated in Figure 10b. The yield stress values were 345 MPa and 400 MPa for steel plates and steel reinforcements, and the elastic modulus was set to 206 GPa [10].

### 4.3. Validation of FE Model

The comparison between the numerical simulation results of the load-slip curves and the results from the push-out test is illustrated in Figure 11. The simulated curve closely aligned with the experimental curves, demonstrating good agreement. Due to the omission of damage consideration in the welded stud material, the deviation between the simulated and experimental curves was minimal in the elastic phase but became more pronounced in the nonlinear phase. These differences were largely attributed to external factors such as sleeve misalignment, grout leakage, welding quality, etc. In general, the numerical model showed reasonable consistency with the tests, effectively reflecting the shear behavior of the connection.

Figure 12 displays the distribution of compressive damage in concrete and the Mises stress distribution in the welded studs at the ultimate state, which shows that the zone of concrete damage in the sleeved stud connection was slightly smaller than that in the conventional welded stud connection. Due to the fact that the damage factor for UHPC was defined only up to 0.7, the color inside the sleeve appeared lighter. In reality, this portion of UHPC has a higher degree of damage, resembling a nearly pure black color, aligning with the failure pattern as observed in Figure 5. Comparing the stress contour plots of the welded studs, the Mises stress contours of the sleeved stud were similar to those of the conventional welded stud. However, the deformation at the base of the welded stud was markedly different; the sleeved stud exhibited a corner folding at its base, and the overall bending deformation was less pronounced than that of the conventional welded stud. This suggested that the sleeved studs experienced less bending-shear coupling, demonstrating superior shear resistance performance.

## 5. Parametric Analysis

### 5.1. Concrete Material

To compare the load-carrying performance between sleeved studs and conventional welded studs, FE models for push-out tests were established using five different concrete materials. Numbering rules follow “base model—material outside sleeve—material inside sleeve” (C represents C50 concrete, U represents UHPC). The load-slip curves for various stud connections are shown in Figure 13. Table 4 summarizes the FE models and the shear performance results, where the shear performance indicators were compared with the conventional welded stud SN12-C used as the baseline.

The computational results revealed that the shear carrying capacity of SS12-C-C was 24% lower than SN12-C, and the shear stiffness decreased by 7% to 8%. Additionally, the shear carrying capacity of SS12-U-U was 10% lower than SN12-U, with only a 2% to 4% reduction in shear stiffness. This revealed that the corrugated metal sleeve weakened the integrity of the concrete block and decreased the shear-carrying capacity of the welded stud. Furthermore, the reduction in shear-carrying capacity and stiffness was more pronounced in C50 concrete. To address the issue of decreased shear-carrying capacity, it is suggested to use the SS12-C-U stud configuration, as seen in the sleeved stud specimens in this push-out test. Grouting the sleeve with UHPC increased the shear-carrying capacity by approximately 8% and enhanced shear stiffness by 22% to 27%, compared with grouting using C50 concrete. This approach utilizes high-performance materials to compensate for inefficiencies due to structural configuration.

In conclusion, it is recommended that in practical bridge design, the interior of the sleeve be grouted using high-strength mortar, or UHPC, with a higher elastic modulus and better flowability, instead of ordinary concrete containing fine aggregates.

### 5.2. Stud Height

FE models with varying stud heights were established, ranging from 80 mm to 250 mm, corresponding to slenderness ratios ranging from 3.6 to 11.4. The numbering rules follows the rule of “base model—stud height”. Figure 14 shows the shear stiffness variation for sleeved stud connectors with different stud heights. Table 5 presents the simulation results for varying stud heights, with the baseline model being SS12-H200.

The numerical results indicated that as the stud height increased, both the shear carry capacity and shear stiffness of the studs exhibited a decreasing trend, although the magnitude of the change was limited. This suggested that the shear behavior of the studs primarily occurred at the base and had limited dependence on the stud height. Previous studies have shown that conventional welded stud connectors are in a bending-shear coupling state during the push-out test, and the bending effect on the stud becomes more pronounced with the increase in the stud height. However, in the current simulation process, the UHPC near the base of the sleeved stud had higher stiffness, preventing the stud from bending. Therefore, the numerical results were insensitive to changes in stud height.

It is also noteworthy that when the stud slenderness ratio was less than 4, the shear performance of the stud significantly improved, with a roughly 10% increase in shear stiffness. When the slenderness ratio was between 5 and 11, the shear performance remained within 4%. This implicitly indicated that for sleeved studs within this range, a unified formula can be used for calculation.

### 5.3. Sleeve Diameter and Stud Diameter

In the push-out test, metal corrugated sleeves with an outer diameter of 60 mm were selected as the sleeves for the welded studs, while the influence of the diameters of the welded studs and sleeves was not studied. Thus, FE models with varying sleeve diameters (ranging from 50 mm to 90 mm) and varying welded stud diameters (19 mm, 22 mm, 25 mm) were designed. The numbering rule follows the rule of “base model—R welded stud diameter—D steel sleeve diameter”, with the results shown in Table 6. The benchmarks are the respective D60 models.

The shear carrying capacity and shear stiffness with changes in steel sleeve diameter for three different welded studs are depicted in Figure 15a,b. When the steel sleeve diameter increased, both the shear strength and shear stiffness of the welded studs increased, indicating that the UHPC inside the sleeve provided stronger resistance. When the steel sleeve diameter exceeded 80 mm, the shear capacity increase slowed down. Although the FE results showed that larger steel sleeve diameters enhanced the shear capacity, mitigating the weakening effect of the metal sleeve. In practical bridge engineering, if the welded studs are arranged in clusters, and considering spatial constraints and the requirements for rebar clearances, the steel sleeve diameter should not exceed 70 mm.

For welded stud connectors with steel sleeve diameters ranging from 50 mm to 70 mm, the shear carrying capacity and shear stiffness varied within ±3%. Within this range, the design of connectors can be made without additional consideration for steel sleeve diameter.

Comparing the shear performance of sleeved stud connectors with different welded stud diameters, in terms of shear carrying capacity, the effect of the steel sleeve diameter was more significant for the larger welded stud diameter. For a welded stud connector with a 25 mm diameter, the variation ranged from −3% to 10%, while for a welded stud connector with a 19 mm diameter, the variation ranged from −3% to 6%. From the perspective of shear stiffness, as the welded stud diameter increased, the shear stiffness was less affected by the steel sleeve diameter. For a welded stud connector with a 25 mm diameter, the variation ranged from −3% to 6%, while for a welded stud connector with a 19 mm diameter, the variation exceeded 10%. From these observations, it is inferred that the shear stiffness of larger diameter welded studs was more closely associated with the material of the welded stud itself. Therefore, the impact of changes in steel sleeve diameter was less significant for larger diameter welded studs.

## 6. Conclusions

This study conducted push-out tests and numerical simulations on a novel sleeved stud connector for steel-concrete composite structures, comparing its shear performance with conventional welded studs. The conclusions drawn are as follows:(1)In the push-out tests, both the sleeved studs and conventional welded studs exhibited shear failure at the root. The concrete failure zone of the sleeved studs was smaller, and the bending deformation at the root of the stud was less than that of conventional welded studs. This indicated that sleeved studs experienced a higher portion of the shear effect and less influence from bending-shear coupling.(2)Although the metal corrugated sleeve of the sleeved studs weakened the overall integrity of the concrete block, this deficiency could be compensated by grouting the sleeve with high-performance concrete materials.(3)The sleeved studs had higher shear-carrying capacity and shear stiffness than conventional welded studs. Besides, the slip at the ultimate load was relatively small. Compared to conventional welded studs, the advantages of sleeved studs in improving shear carrying capacity were insignificant, with an increase of 4% to 8%. However, sleeved studs exhibited stronger resistance against slip deformation, with shear stiffness improving by 25% to 35%.(4)Since the shear behavior of sleeved studs mainly occurred at the root, the influence of stud height on shear performance was minimal. However, sleeve diameter and welded stud diameter had a great impact on shear carrying capacity and shear stiffness. As the sleeve diameter varied from 50 mm to 100 mm, shear carrying capacity and shear stiffness increased by 8% and 10%, respectively, because the UHPC near the root of the welded studs effectively enhanced the shear carrying capacity of the connector.

## Figures and Tables

**Figure 1 materials-17-03326-f001:**
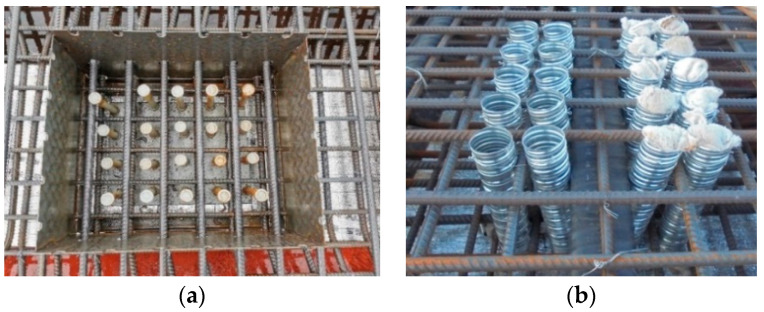
(**a**) Photos of the encased cluster stud connector. (**b**) Photos of the sleeved cluster stud connector.

**Figure 2 materials-17-03326-f002:**
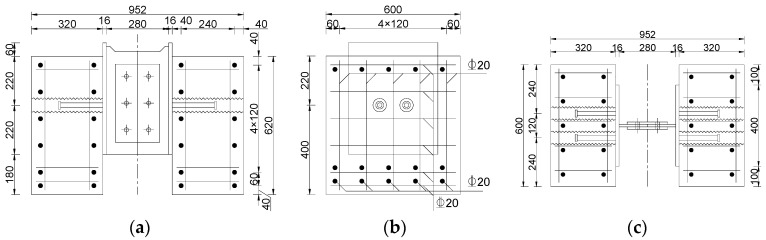
Configurations of specimen SS12 (mm). (**a**) Front view. (**b**) Side view. (**c**) Top view.

**Figure 3 materials-17-03326-f003:**
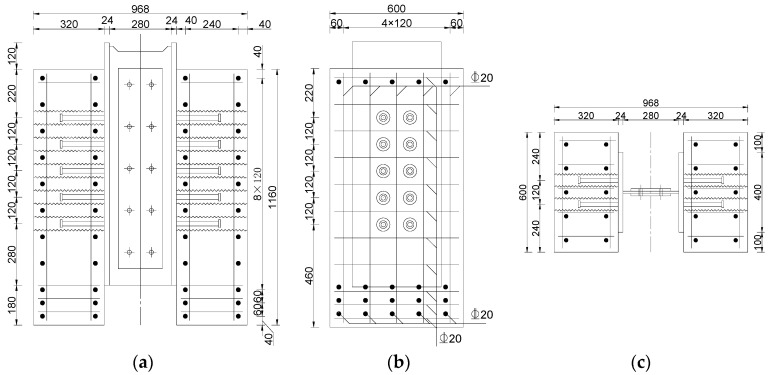
Configurations of specimen SSL52 (mm). (**a**) Front view. (**b**) Side view. (**c**) Top view.

**Figure 4 materials-17-03326-f004:**
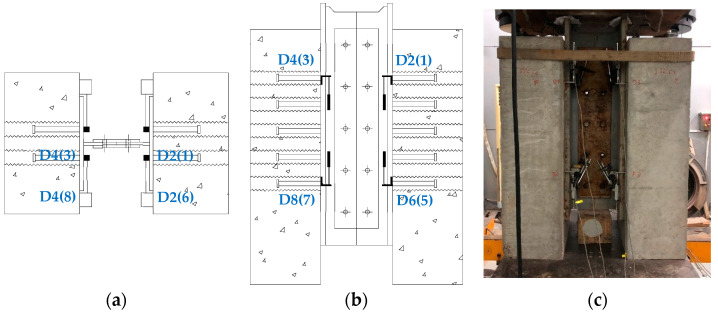
Measurement point layout and loading device. (**a**) Layout of measurement points for specimens with multiple rows of studs (plan). (**b**) Layout of measurement points for specimens with multiple rows of studs (front). (**c**) Loading device.

**Figure 5 materials-17-03326-f005:**
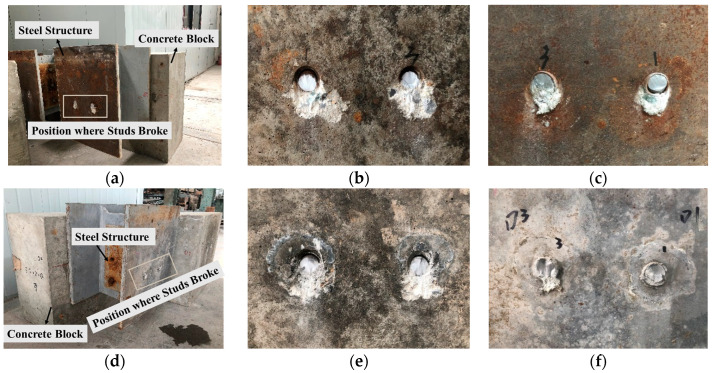
Failure modes of stud connector specimens. (**a**) Specimen SN12-A. (**b**) Concrete side (SN12-A). (**c**) Steel side (SN12-A). (**d**) Specimen SS12-A. (**e**) Concrete side (SS12-A). (**f**) Steel side (SS12-A).

**Figure 6 materials-17-03326-f006:**
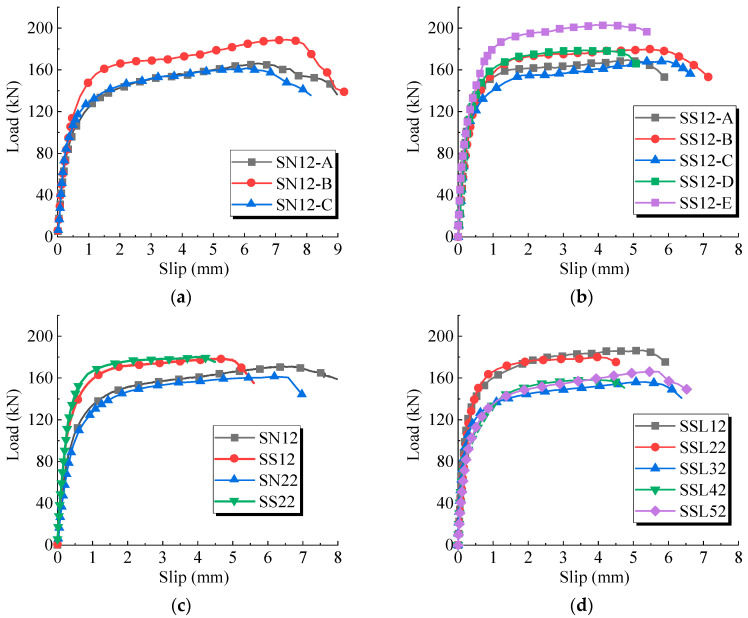
Load–slip curves in the push-out tests. (**a**) SN12. (**b**) SS12. (**c**) SN22 and SS22. (**d**) SSL group.

**Figure 7 materials-17-03326-f007:**
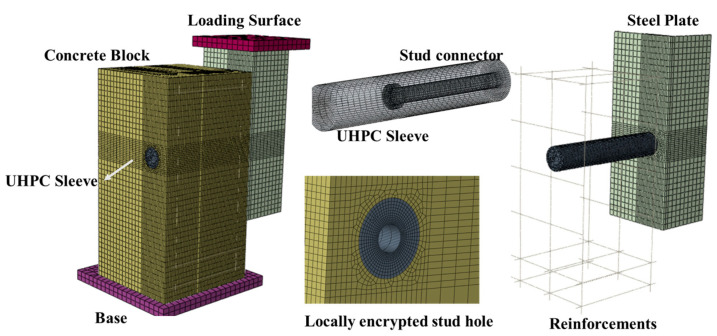
FE model of push-out test.

**Figure 8 materials-17-03326-f008:**
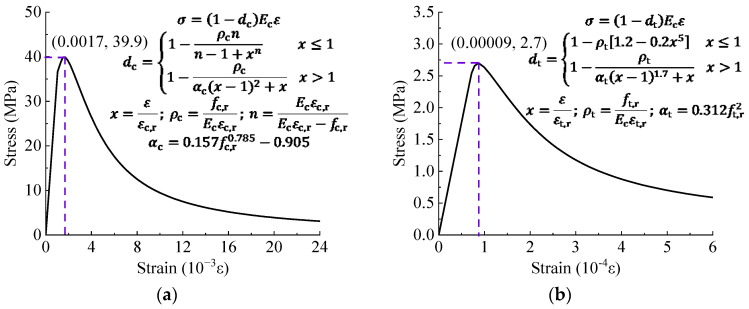
Constitutive models of C50 ordinary concrete. (**a**) Compressive stress; (**b**) Tensile stress.

**Figure 9 materials-17-03326-f009:**
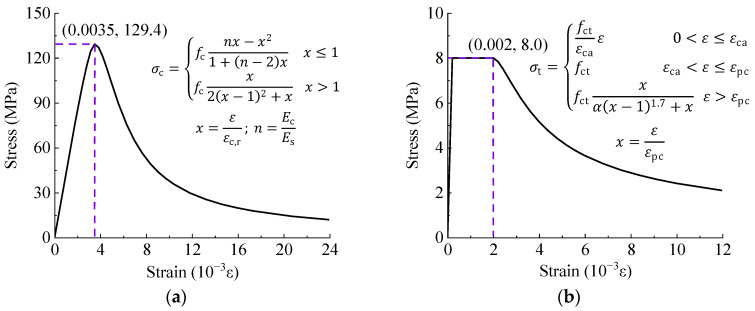
Constitutive model of UHPC. (**a**) Compressive stress; (**b**) Tensile stress.

**Figure 10 materials-17-03326-f010:**
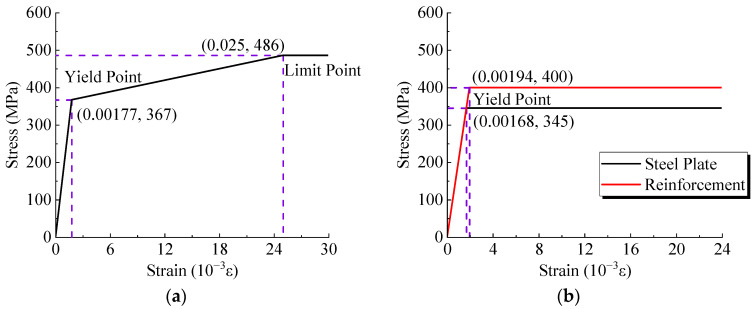
Constitutive models of steel. (**a**) Welded studs. (**b**) Steel plates and reinforcements.

**Figure 11 materials-17-03326-f011:**
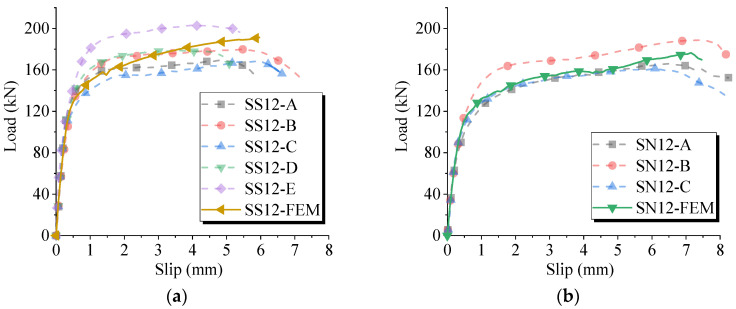
Comparison of numerical and experimental load–deflection curves. (**a**) Comparison for sleeved stud connector. (**b**) Comparison for conventional welded stud.

**Figure 12 materials-17-03326-f012:**
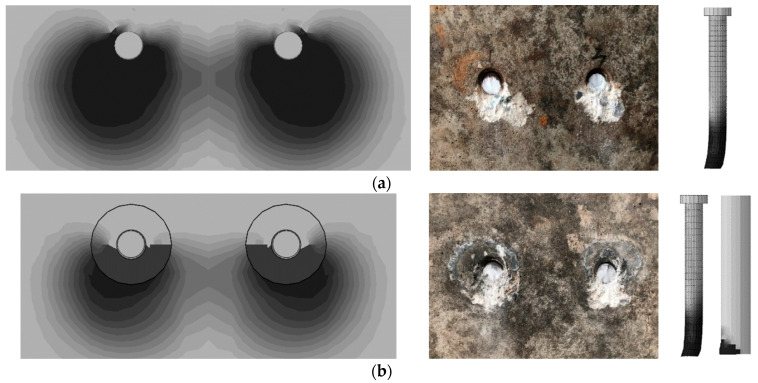
Comparison of failure modes between FE simulation and experimental results. (**a**) SN12. (**b**) SS12.

**Figure 13 materials-17-03326-f013:**
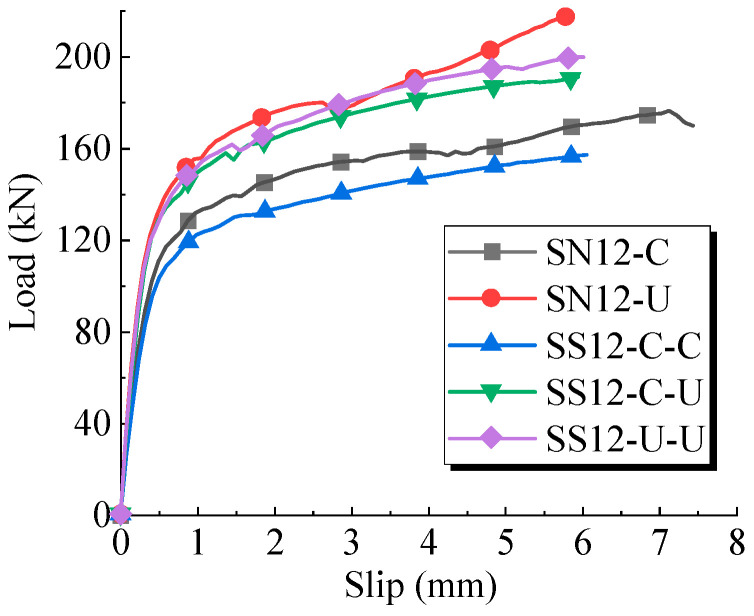
Load slip curves of stud connectors with different construction forms.

**Figure 14 materials-17-03326-f014:**
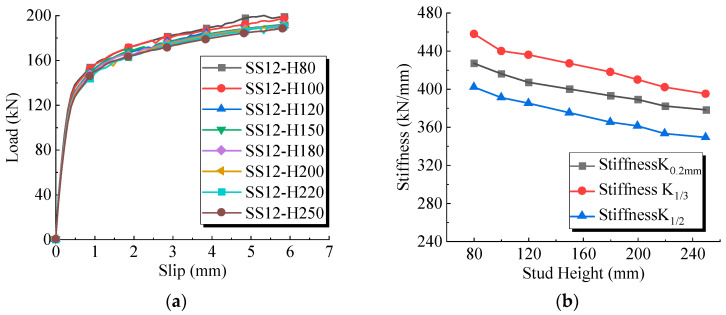
(**a**) Load slip curves of welded studs with different heights. (**b**) The influence of height variation of welded studs on shear stiffness.

**Figure 15 materials-17-03326-f015:**
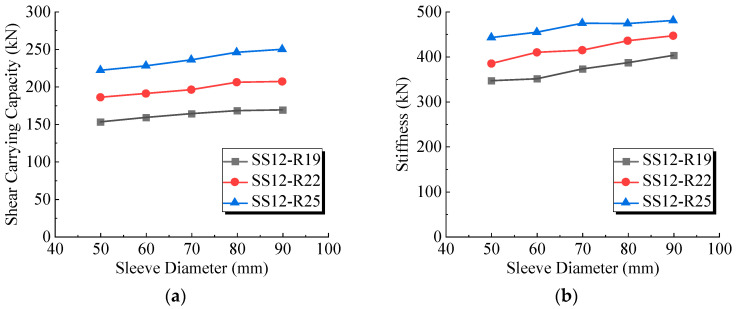
(**a**) Variation of ultimate shear bearing capacity with sleeve diameter (**b**) Variation of shear stiffness $K_{1/3}$ with sleeve diameter.

**Table 1 materials-17-03326-t001:** Groups of test specimens.

Specimen Label	Stud Size Diameter × Height (mm)	Layout of Studs (Each Side)	Sleeve Size Diameter × Height (cm)	Grouting Material in Sleeve	Specimen Number
SN12	22 × 200	1 × 2	6 × 32	None	3
SS12		1 × 2		UHPC	5
SN22		2 × 2		None	1
SS22		2 × 2		UHPC	1
SSL12		1 × 2		UHPC	1
SSL22		2 × 2		UHPC	1
SSL32		3 × 2		UHPC	1
SSL42		4 × 2		UHPC	1
SSL52		5 × 2		UHPC	1

**Table 2 materials-17-03326-t002:** Material properties.

Concrete Properties	Cubic Compressive Strength (MPa)	Axial Compressive Strength (MPa)	Elastic Modulus (GPa)
C50 concrete	53.2	39.9	35.77
UHPC	136.8	129.2	44.00
**Stud Properties**	**Yield Strength (MPa)**	**Tensile Strength (MPa)**	**Elastic Modulus (GPa)**
ML15 cylindrical head stud	367	486	207

**Table 3 materials-17-03326-t003:** Summarization of push-out test results.

Group	Numbering	$\mathrm{Shear}\;\mathrm{Carrying}\;\mathrm{Capacity}\;V_{u}$ (kN)		$\mathrm{Slip}\;\mathrm{at}\;\mathrm{Shear}\;\mathrm{Carrying}\;\mathrm{Capacity}S_{p}$ (mm)		$\mathrm{Shear}\;\mathrm{Stiffness}\;K_{0.2\;mm}$ (kN/mm)		$\mathrm{Shear}\;\mathrm{Stiffness}\;K_{1/3}$ (kN/mm)		$\mathrm{Shear}\;\mathrm{Stiffness}\;K_{1/2}$ (kN/mm)	
SN12	A	166.8	Avg: 172.6 0%	6.45	Avg: 6.67 0%	312	Avg: 329 0%	327	Avg: 333 0%	247	Avg: 272 0%
	B	189.7		7.45		331		321		266	
	C	161.3		6.90		342		351		313	
SS12	A	169.7	Avg: 180.3 ↑4.5%	4.59	Avg: 4.82 ↓27.7%	444	Avg: 434 ↑31.9%	422	Avg: 462 ↑38.7%	429	Avg: 402 ↑47.8%
	B	180.0		5.09		370		387		331	
	C	168.7		5.90		455		457		446	
	D	179.1		4.20		424		456		398	
	E	203.8		4.34		477		629		434	
SN22		161.2	↓6.6%	6.21	↓6.9%	275	↓16.4%	270	↓18.9%	233	↓14.3%
SS22		180.6	↑4.6%	3.81	↓42.9%	446	↑35.6%	485	↑45.6%	432	↑58.8%
SSL12		186.6	0%	6.08	0%	513	0%	641	0%	527	0%
SSL22		179.4	↓3.8%	6.10	↑0.3%	485	↓5.5%	574	↓10.4%	501	↓4.9%
SSL32		156.4	↓16.2%	6.51	↑7.1%	458	↓11.1%	714	↑11.3%	525	↓3.8%
SSL42		158.3	↓15.2%	5.02	↓17.4%	406	↓20.9%	606	↓6%	408	↓22.6%
SSL52		166.4	↓10.8%	7.77	↑27.8%	385	↓25.0%	451	↓29.6%	350	↓33.6%

**Table 4 materials-17-03326-t004:** Influence of concrete material on sleeved stud connectors.

Model Numbering	Material		$V_{u}$ (kN)		$K_{0.2\;mm}$ (kN/mm)		$K_{1/3}$ (kN/mm)		$K_{1/2}$ (kN/mm)	
	Outside Sleeve	Inside Sleeve								
SN12-C	C50	N.A.	176	0%	318	0%	323	0%	284	0%
SN12-U	UHPC	N.A.	218	↑24%	424	↑33%	445	↑38%	367	↑29%
SS12-C-C	C50	C50	152	↓14%	291	↓8%	299	↓7%	268	↓6%
SS12-C-U	C50	UHPC	191	↑8%	389	↑22%	410	↑27%	361	↑27%
SS12-U-U	UHPC	UHPC	200	↑14%	411	↑29%	438	↑36%	372	↑31%

N.A.: There is no sleeve.

**Table 5 materials-17-03326-t005:** Influence of stud height on sleeved stud connectors.

Model Numbering	Stud Height (mm)	$V_{u}$ (kN)		$K_{0.2\;mm}$ (kN/mm)		$K_{1/3}$ (kN/mm)		$K_{1/2}$ (kN/mm)	
SS12-H80	80	199	↑4%	427	↑10%	458	↑12%	402	↑11%
SS12-H100	100	198	↑4%	416	↑7%	440	↑7%	391	↑8%
SS12-H120	120	191	0%	407	↑5%	436	↑6%	385	↑7%
SS12-H150	150	192	↑1%	400	↑3%	427	↑4%	375	↑4%
SS12-H180	180	192	↑1%	393	↑2%	418	↑2%	365	↑1%
SS12-H200	200	191	0%	389	0%	410	0%	361	0%
SS12-H220	220	190	↓1%	382	↓2%	402	↓2%	353	↓2%
SS12-H250	250	189	↓1%	378	↓3%	395	↓4%	349	↓3%

**Table 6 materials-17-03326-t006:** Influence of sleeve diameter and stud diameter on the shear performance of sleeved stud connectors.

Model Numbering	Stud Diameter/Sleeve Diameter	$V_{u}$ (kN)		$K_{0.2\;mm}$ (kN/mm)		$K_{1/3}$ (kN/mm)		$K_{1/2}$ (kN/mm)	
SS12-R19-D50	0.38	153	↓3%	322	↓3%	347	↓1%	302	↓2%
SS12-R19-D60	0.32	159	0%	331	0%	351	0%	307	0%
SS12-R19-D70	0.27	164	↑3%	344	↑4%	373	↑6%	318	↑4%
SS12-R19-D80	0.24	168	↑6%	352	↑6%	387	↑10%	318	↑4%
SS12-R19-D90	0.21	169	↑6%	362	↑9%	403	↑15%	331	↑7%

SS12-R22-D50	0.44	186	↓3%	374	↓4%	385	↓6%	349	↓3%
SS12-R22-D60	0.37	191	0%	389	0%	410	0%	361	0%
SS12-R22-D70	0.31	196	↑3%	398	↑2%	415	↑1%	369	↑2%
SS12-R22-D80	0.28	206	↑8%	413	↑6%	436	↑6%	374	↑4%
SS12-R22-D90	0.24	207	↑8%	416	↑7%	447	↑9%	380	↑5%

SS12-R25-D50	0.5	222	↓3%	431	↓3%	443	↓3%	402	↓3%
SS12-R25-D60	0.42	228	0%	443	0%	455	0%	414	0%
SS12-R25-D70	0.35	236	↑4%	456	↑3%	475	↑4%	426	↑3%
SS12-R25-D80	0.31	246	↑8%	458	↑3%	474	↑4%	423	↑2%
SS12-R25-D90	0.28	250	↑10%	469	↑6%	481	↑6%	430	↑4%

## Data Availability

The original contributions presented in the study are included in the article, further inquiries can be directed to the corresponding author.

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
