# Peer review of "Shear Performance Study of Sleeved Stud Connectors in Continuous Composite Girder"

_materials, 2024, doi:10.3390/ma17133326_

Round 1
Reviewer 1 Report
Comments and Suggestions for Authors
Dear authors,
The only problem of greater significance that I see is the issue of poor mesh selection for calculations. Not only is the mesh poorly chosen in the area of the hole, but it is additionally different for the cases shown (Figure 13). The influence of the mesh on the results should be determined.
Moreover:
- Figure 4 adds nothing new to the article and should be removed
- lack of assumed value of friction coefficient and information on what basis it was adopted
- no information about the integration scheme used in the FEM and its parameters
- Fig. 6a and 6d are not very intelligible and should be presented in another way
- too often sentences start with the word Figure
- in the case of the description of the material models, I miss a few sentences about the fact that samples for material testing were prepared in advance according to certain standards.
- Fig. 13 lacks a stud connector after research
Author Response
The only problem of greater significance that I see is the issue of poor mesh selection for calculations. Not only is the mesh poorly chosen in the area of the hole, but it is additionally different for the cases shown (Figure 13). The influence of the mesh on the results should be determined.
Response: Thanks for the reviewer’s instructive advice. Actually, the modeling method is carefully determined after many attempts, considering both efficiency and accuracy. Adopting the reviewer’s advice, we optimize the mesh work as shown in Figure 7 (both concrete and steel part are optimized). The final results are similar to the original one. Thus, we consider the numerical analysis accurate.
As for the difference between Figure 12 and Figure 7, actually Figure 7 doesn’t show the mesh condition around the stud hole, but only the mesh condition of the UHPC Sleeve. Besides, Figure 12 is jointed by two similar pictures intercepted from the model to be compared with the test.
We have adopted the reviewer’s advice and revised the modeling methods in Figure 7 and the numerical results in figure 12. As the results between the two kind of model shows few differences, the other numerical results are not revised. In the future research, we will conduct the mesh work more carefully according to the reviewer’s advice.
- Figure 4 adds nothing new to the article and should be removed.
Response: We have removed Figure 4 in section 2.1 in the revised manuscript.
- Lack of assumed value of friction coefficient and information on what basis it was adopted.
Response: We have added the illustration of the friction coefficient with the reference in section 4.1 in the revised manuscript.
- No information about the integration scheme used in the FEM and its parameters.
Response: The simulation employed Dynamic/Explicit method for analysis, which integrates motion equations by the central difference method without solving iterative equation. The target time increment of the mass scaling was set as 0.0001 to improve the calculation efficiency. We have added the illustration in section 4.1 in the revised manuscript.
- Fig. 6a and 6d are not very intelligible and should be presented in another way.
Response: We have added illustration in the figures to make them intelligible in section 4.1 in the revised manuscript.
- Too often sentences start with the word Figure.
Response: We have optimized these sentences in the revised manuscript, please refer to section 3.1, 3.2, 4.3, 4.4.
- Fig. 13 lacks a stud connector after research
Response: As the Ultra-High Performance Concrete is hard to destroy, we didn’t get the stud connector out of the concrete block. The failure mood of the stud connector can be inferred through the exposed part of the stud root and the FEM results. Thanks for the reviewer’s suggestion, we will try to get the stud connector out of the concrete block for our next research.
Reviewer 2 Report
Comments and Suggestions for Authors
The topic is within the scope of the Journal and experiments are, based on my expertise, consistent. these are my comments:
The captions of figures 2 and 3 are missing or, probably, the same figure was wrongly inserted in the manuscript two times.
The authors should better explain why the influence of stud height on shear performance was minimal.
References are not in the correct format.
Moreover, the bibliography is not adequated in my opinion; it should be widely extended and, based on this, the initial state of the art should be enriched.
Comments on the Quality of English LanguageEnglish language is suitable for publication.
Author Response
We sincerely appreciate the valuable comments, suggestions and questions from the reviewer. The manuscript has been revised accordingly, in which the changes have been highlighted for the review convenience. Meanwhile, the point-to-point responses to the reviewer's comments are listed as follows.
The topic is within the scope of the Journal and experiments are, based on my expertise, consistent. these are my comments.
Response: Thanks a lot for the reviewer’s advices. All the comments and suggestions have been carefully considered and revised in the resubmitted manuscript.
- The captions of figures 2 and 3 are missing or, probably, the same figure was wrongly inserted in the manuscript two times.
Response: We have revised the captions of Figure 2 and Figure 3 in Section 2.1.
- The authors should better explain why the influence of stud height on shear performance was minimal.
Response: During the testing process, the sleeved studs primarily underwent direct shear at the base, while the conventional welded studs exhibited more pronounced bending effects. As the major loading position of the sleeved stud is at the root, the influence of stud height on shear performance was minimal. We have added the illustration to Section 3.1 in the revised manuscript.
- References are not in the correct format.
Response: We have checked and revised the references.
- Moreover, the bibliography is not adequated in my opinion; it should be widely extended and, based on this, the initial state of the art should be enriched.
Response: We have extended the bibliography and enriched the state of the art in the revised manuscript.
Reviewer 3 Report
Comments and Suggestions for Authors
The manuscript presents tests of connecting a reinforced concrete slab with a steel girder using studs. These types of connections are standard in steel-concrete composite structures. The authors tested a modification of the conventional solution consisting in the introduction of UHPC concrete around the stud. The research was preceded by an analysis of similar solutions, described in the introduction. The lay-out of models for testing and the scope of the research performed were clearly described. I also have no significant comments regarding the discussion of the results. They honestly present the advantages and disadvantages of the proposed solution.
1) I did not find any information on material testing of reinforcement and steel plates. Only the model in Figure 11b is shown.
2) If the stress-strain relationship was determined in the full load spectrum in steel tests, it can be shown as a background in Figure 11.
3) Section 2.2 Material properties discusses the results of material tests, it would be advisable to provide standards according to. which the tests were performed, or at least the size of the concrete samples.
4) The modulus of elasticity in Table 2.2 is given as (x103 MPa), I suggest changing it to GPa or writing it with the index 103.
5) There is a wrong unit in line 138 - stud diameter is 22mm.
Author Response
We sincerely appreciate the valuable comments, suggestions and questions from the reviewer. The manuscript has been revised accordingly, in which the changes have been highlighted for the review convenience. Meanwhile, the point-to-point responses to the reviewer's comments are listed as follows.
The manuscript presents tests of connecting a reinforced concrete slab with a steel girder using studs. These types of connections are standard in steel-concrete composite structures. The authors tested a modification of the conventional solution consisting in the introduction of UHPC concrete around the stud. The research was preceded by an analysis of similar solutions, described in the introduction. The lay-out of models for testing and the scope of the research performed were clearly described. I also have no significant comments regarding the discussion of the results. They honestly present the advantages and disadvantages of the proposed solution.
Response: Thanks a lot for the reviewer’s instructive advices. All the comments and suggestions have been carefully considered and revised in the resubmitted manuscript.
- I did not find any information on material testing of reinforcement and steel plates. Only the model in Figure 11b is shown.
Response: As the material properties of the steel plates (where the studs are welded) and the reinforcement in the concrete block have minimal influence on push-out tests, we only conducted material tests on studs and concrete. We have added the reference for Figure 11b to Section 4.2 in the revised manuscript.
- If the stress-strain relationship was determined in the full load spectrum in steel tests, it can be shown as a background in Figure 11.
Response: Thanks to the reviewer for the advice. As the full load spectrum of the stress-strain relationship do not have influence on the constitutive model, we think the figure is clearer using this version.
- Section 2.2 Material properties discusses the results of material tests, it would be advisable to provide standards according to. which the tests were performed, or at least the size of the concrete samples.
Response: We have added the standards to Section 2.2 in the revised manuscript.
- The modulus of elasticity in Table 2.2 is given as (x103 MPa), I suggest changing it to GPa or writing it with the index 103.
Response: We have revised the modulus of elasticity unit in Table 2 to GPa in the revised manuscript.
- There is a wrong unit in line 138 - stud diameter is 22mm.
Response: We have revised the stud diameter to 22mm in section 2.1 in the revised manuscript.
Round 2
Reviewer 1 Report
Comments and Suggestions for Authors
Dear authors,
I did not find changes in the article regarding most of my earlier questions and suggestions from the review, namely:
The only problem of greater significance that I see is the issue of poor mesh selection for calculations. Not only is the mesh poorly chosen in the area of the hole, but it is additionally different for the cases shown (Figure 13). The influence of the mesh on the results should be determined.
Moreover:
- Figure 4 adds nothing new to the article and should be removed
- lack of assumed value of friction coefficient and information on what basis it was adopted
- no information about the integration scheme used in the FEM and its parameters
- Fig. 6a and 6d are not very intelligible and should be presented in another way
- too often sentences start with the word Figure
- Fig. 13 lacks a stud connector after research
Author Response
I did not find changes in the article regarding most of my earlier questions and suggestions from the review.
Response: Thanks a lot for the reviewer’s instructive advices and sorry for the delayed response. Last time when I submitted the revised manuscript, there were only comments from two reviewers. Thus, the manuscript was revised according to the comments from two reviewers. In this resubmitted manuscript, all of your comments and suggestions have been carefully considered and revised.
The only problem of greater significance that I see is the issue of poor mesh selection for calculations. Not only is the mesh poorly chosen in the area of the hole, but it is additionally different for the cases shown (Figure 13). The influence of the mesh on the results should be determined.
Response: Thanks for the reviewer’s instructive advice. Actually, the modeling method is carefully determined after many attempts, considering both efficiency and accuracy. Adopting the reviewer’s advice, we optimize the mesh work as shown in Figure 7 (both concrete and steel part are optimized). The final results are similar to the original one. Thus, we consider the numerical analysis accurate.
As for the difference between Figure 12 and Figure 7, actually Figure 7 doesn’t show the mesh condition around the stud hole, but only the mesh condition of the UHPC Sleeve. Besides, Figure 12 is jointed by two similar pictures intercepted from the model to be compared with the test.
We have adopted the reviewer’s advice and revised the modeling methods in Figure 7 and the numerical results in figure 12. As the results between the two kind of model shows few differences, the other numerical results are not revised. In the future research, we will conduct the mesh work more carefully according to the reviewer’s advice.
- Figure 4 adds nothing new to the article and should be removed.
Response: We have removed Figure 4 in section 2.1 in the revised manuscript.
- Lack of assumed value of friction coefficient and information on what basis it was adopted.
Response: We have added the illustration of the friction coefficient with the reference in section 4.1 in the revised manuscript.
- No information about the integration scheme used in the FEM and its parameters.
Response: The simulation employed Dynamic/Explicit method for analysis, which integrates motion equations by the central difference method without solving iterative equation. The target time increment of the mass scaling was set as 0.0001 to improve the calculation efficiency. We have added the illustration in section 4.1 in the revised manuscript.
- Fig. 6a and 6d are not very intelligible and should be presented in another way.
Response: We have added illustration in the figures to make them intelligible in section 4.1 in the revised manuscript.
- Too often sentences start with the word Figure.
Response: We have optimized these sentences in the revised manuscript, please refer to section 3.1, 3.2, 4.3, 4.4.
- Fig. 13 lacks a stud connector after research
Response: As the Ultra-High Performance Concrete is hard to destroy, we didn’t get the stud connector out of the concrete block. The failure mood of the stud connector can be inferred through the exposed part of the stud root and the FEM results. Thanks for the reviewer’s suggestion, we will try to get the stud connector out of the concrete block for our next research.
Round 3
Reviewer 1 Report
Comments and Suggestions for Authors
Dear authors,
I wish you continued success.
Only one comment remains:
there should be a space between the value and the ed unit (l.144 and others) - is "20mm" and should be "20 mm".